# Applicability and Limitations of a Capillary-LC Column-Switching System Using Hybrid Graphene-Based Stationary Phases

**DOI:** 10.3390/molecules28134999

**Published:** 2023-06-26

**Authors:** João Victor Basolli Borsatto, Edvaldo Vasconcelos Soares Maciel, Alejandro Cifuentes, Fernando Mauro Lanças

**Affiliations:** 1Laboratory of Chromatography, Institute of Chemistry at Sao Carlos, University of Sao Paulo, P.O. Box 780, Sao Carlos 13566590, Brazil; jvictorborsatto@usp.br (J.V.B.B.); daltoniqsc@gmail.com (E.V.S.M.); 2Laboratory of Foodomics, Institute of Food Science Research (CIAL, CSIC), Nicolás Cabrera 9, 28049 Madrid, Spain; 3Clemens Schöpf Institute, Department of Chemistry, Technical University of Darmstadt, 64287 Darmstadt, Germany

**Keywords:** capillary liquid chromatography, graphene-based materials, packed column, column switching, direct injection, SiGO, SiGO-C18ec

## Abstract

Graphene oxide sheets fixed over silica particles (SiGO) and their modification functionalized with C18 and endcapped (SiGO-C18ec) have been reported as sorbents for extraction and analytical columns in LC. In this study, a SiGO column was selected as the extraction column and a SiGO-C18ec as the analytical column to study the applicability and limitations of a column-switching system composed exclusively of columns packed with graphene-based sorbents. Pyriproxyfen and abamectin B1a were selected as the analytes, and orange-flavored carbonated soft drinks as the matrix. The proposed system could be successfully applied to the pyriproxyfen analysis in a concentration range between 0.5 to 25 µg/mL presenting a linearity of R^2^ = 0.9931 and an intra-day and inter-day accuracy of 82.2–111.4% (RSD < 13.3%) and 95.5–99.8% (RSD < 12.7%), respectively. Furthermore, the matrix composition affected the area observed for the pyriproxyfen: the higher the concentration of orange juice in the soft drink, the higher the pyriproxyfen the signal observed. Additionally, the SiGO extraction column presented a life use of 120 injections for this matrix. In contrast, the proposed system could not apply to the analysis of abamectin B1a, and the SiGO-C18ec analytical column presented significant tailing compared to a similar approach with a C18 analytical column.

## 1. Introduction

Graphene and graphene-based materials are used in diverse analytical chemistry applications [1,2,3]. These materials have also been receiving increasing attention in sample preparation and liquid chromatography (LC) in the last decade. Graphene-based phases can be present in different ways. Fe_3_O_4_ magnetic particles containing graphene [4] or graphene oxide [5], graphene-based aerogels [6,7] and hydrogels [8,9], and graphene-based material fixed on silica particles [10,11,12] are examples of how these materials can be produced. Other possibilities of application for graphene-based materials are summarized in recent reviews [13,14,15].

Applications in separation science are as diverse as the possibilities of graphene-based material compositions [16,17,18]. Techniques such as stir bar sorptive extraction [16,19], solid-phase extraction [10,20], fiber solid-phase microextraction [20,21], in-tube solid-phase microextraction [12,22], and others [1,23,24] are examples of graphene-based material applications in sample preparation techniques. A particular focus can be given to column-switching methods coupled with LC or miniaturized LC because they minimize the steps of sample preparation procedures [25,26]. Sample preparation errors are one of the three most predominant sources of error in analytical chemistry [27]. Among the graphene-based phases applied to column-switching, graphene oxide sheets fixed to amino-silica particles (SiGO) and graphene oxide sheets fixed to amino-silica particles functionalized with C18 and endcapped (SiGO-C18ec) can be highlighted. These phases have been successfully applied in environmental, biological, and food analysis [28,29]. For example, extraction columns (also known as pre-concentration columns) packed with SiGO-C18ec particles have been applied with success in studies of pesticides in sugarcane spirits [30], mycotoxin separation in beverages [31], and xanthines in coffee beverages [32]. Examples of their application in a column-switching setup containing SiGO particles in the extraction column are the determination of β-lactams from environmental water samples [33], pesticides in sugarcane spirits [30], and resolution of antidepressant and antiepileptic drugs in urine [34].

Graphene-based materials have also been explored as packing material for analytical columns in LC [28]. Graphene sheets fixed to silica particles [35,36], graphene oxide sheets fixed to silica particles functionalized with C18 [37,38], graphene sheets fixed to silica particles modified with gold nanoparticles [39], and cellulose-coated reduced oxide sheets fixed to silica particles [40] are examples of some materials that have been applied as stationary phases in liquid chromatography. Columns packed with the SiGO-C18ec phase have been successfully used in quantitative analyses of hormones in urine samples [41]. A recent study compared a SiGO-C18ec with a commercial C18 column, both packed capillary LC columns operating in reverse-phase LC [42]. It was observed that SiGO-C18ec phases present different selectivity from C18 phases. Once graphene-based columns are applicable as both extraction and analytical columns in column-switching mode, it is possible to presume that a column-switching system composed of only graphene-based columns could be viable. This work evaluates the applicability and limitations of a column-switching system consisting exclusively of columns packed with graphene-based stationary phases. To the best of our knowledge, no work has described the use of a column-switching system composed exclusively of graphene-based packed columns in any application, and this a vacancy to be investigated. For this study, a SiGO column was selected as the extraction column and a SiGO-C18ec as the analytical column. In addition, pyriproxyfen and abamectin B1a were selected as analytes, and orange-flavored carbonated soft drinks as a matrix. Orange soft drink, are an interesting matrix to evaluate this proposed column-switching system. Though this matrix is complex and presents solid material in suspension, it can also be directly injected in the column-switching system without pretreatment. Pyriproxyfen and abamectin B1a are insecticides employed to control several pest species in diverse cultures. They have been selected as model analytes for this study because quantitative analysis has already been reported in the literature for both compounds in orange and orange by-product samples [43,44]. The leading figures of merit were evaluated, including linearity, accuracy, and precision of the calibration curve, the matrix effect, and the column life-use parameters. A comparison of the peak shape with a C18 analytical column is presented.

## 2. Results and Discussion

### 2.1. Selection of the Analytes

Pyriproxyfen and abamectin B1a are pesticides from two distinct chemical classes, which present different structures (Figure 1).

Pyriproxyfen mainly comprises aromatic rings and abamectin B1a by aliphatic chains (Figure 1A). Despite these structural differences, pyriproxyfen and abamectin B1a presented closed retention times in the column-switching system composed of the columns packed with SiGO and SiGO-C18ec (Figure 1B). The similarity in the retention time indicated that both analytes were similarly retained in the proposed column-switching system. This characteristic was important to evaluate whether the proposed column-switching system, composed exclusively of columns packed with graphene-based phases, is multipurpose or not. If the system demonstrated successful quantitative analysis for only one of the analytes, it would suggest its limitations as a multipurpose system. Conversely, if it provided suitable quantitative analysis for both compounds, further investigation would be necessary to confirm the multipurpose capabilities of the system. Because of the characteristics mentioned above, these compounds were selected for this study. In addition, the gradient method was optimized to ensure the elution of the monitored compounds and to elute non-monitored compounds that could stay retained in the columns and reduce the life use of the devices (Section 2.6 details the life-use aspects of these columns). Figure 1B depicts two other pieces of evidence: (i) the pyriproxyfen MRM chromatogram presents a more intense signal than the abamectin 1Ba MRM chromatogram; and (ii) the pyriproxyfen MRM chromatogram presents a more pronounced tail, which is almost not present in the abamectin B1a chromatogram.

### 2.2. Loading Method

The first variable evaluated for this study was the loading flow rate. The loading flow rate directly affects the extraction of the analytes in the extraction column. Figure 2 shows the area and the relative standard deviation (% RSD) obtained for both evaluated compounds at 25, 50, and 100 µL/min loading flow rates.

The loading flow rate did not affect the area obtained for the pyriproxyfen (Figure 2A), but a reduction in the % RSD was observed with the decrease in the loading flow rate (Figure 2B). For the abamectin B1a, the 50 µL/min loading flow rate resulted in a higher area (Figure 2C) and lower relative % RSD (Figure 2D) than the other explored. Therefore, the 50 µL/min loading flow rate was selected to be used this study because it allowed for better extraction of both analytes from the matrix.

### 2.3. Linearity of the Calibration Curve

Calibration curves are the core of quantitative analysis [45]. The experimental data in Figure 3 shows that the column-switching method presented an excellent linearity for the study of pyriproxyfen, with R^2^ > 0.99 (Figure 3A).

The % RSD for the pyriproxyfen analysis was lower than 12% for all the concentrations (Figure 3B). Both observations suggest that the experimental setup composed of a first column containing SiGO (extraction column) and a second column of SiGO-C18ec (analytical column) is suitable for analyzing pyriproxyfen in beverages consisting of orange juice. On the other hand, for abamectin B1a, the calibration curve did not present good linearity, with R^2^ < 0.47 (Figure 3C), and gave high values for % RSD (Figure 3D). Based on that observation, it is possible to point out one of the limitations of the explored system: SiGO columns are not multipurpose, at least not at this point of development, and their application could still be limited. These limitations are already expected once graphene-derived columns are under early development. Usually, each kind of reverse-phase stationary phase has a specific range of operations. In some cases, it can be broad (C18, as an example) but limited in others (chiral phases, as an example). Although it demands much more investigation before a conclusion can be drawn, it seems that the graphene-based stationary phase can become one option for specific applications instead of a multipurpose phase.

### 2.4. Intra-Day and Inter-Day Accuracy and Precision

Accuracy and precision are other critical factors in quantitative analysis. For this work, the acceptance criteria were accuracy between 70–110% and precision with % RSD <20%; these values were based on previous work using graphene-based extraction or analytical columns [33,41]. Table 1 presents the accuracy and precision data obtained. For the pyriproxyfen analysis, the intra-day % RSD was lower than 20% for all the points and, in most determinations, lower than 10%.

The inter-day % RSD for this compound was 12.7%, 3.9%, and 2.9% for 2, 8, and 20 µg/mL concentrations, respectively. The intra-day accuracy for the analysis of the pyriproxyfen was between 82.2% and 111.4%, and the inter-day accuracy was between 95.5% and 99.8%. For the analysis of abamectin B1a, the proposed column-switching system was demonstrated not to be reliable. For this compound, obtaining intra-day and inter-day % RSD higher than 20% and accuracy in a wide range between 5.8% and 207% was typical. The observations stated in the last section are confirmed here. Graphene-based stationary phases are promising phases that can be used in a column-switching system, but their applications might not be multipurpose and present limitations.

### 2.5. Matrix Effects

The matrix composition might affect the analytical signal (peak area, as an example) observed, interfering with the analysis. This is called the “matrix effect”. Knowledge about the matrix effect is essential in most analysts’ chromatography column-switching approach to extraction, especially when a sample pretreatment is not included in the analytical workflow before the injection step. To conduct this experiment, two commercial orange-flavored soft drinks with different proportions of orange juice in their compositions and spiked with pyriproxyfen in water were explored. Only pyriproxyfen was used to study the effect of the matrix. In the concentration of 2 µg/mL, the impact of the matrix was minimal. However, the analytes might have been better extracted in orange juice samples than pure water (Figure 4A).

The same trend was observed for the concentration of 8 µg/mL: the extraction in both samples containing orange juice was similar, and both were better than the aqueous sample (Figure 4B). For the 20 µg/mL solution, it is noted that the higher the percentage of orange juice in the matrix composition, the higher the area of the pyriproxyfen obtained (Figure 4C). These observations allow us to conclude that the matrix affects the extraction capability of the SiGO column, and this effect is more perceptible as the concentration of the analytes increases. These observations are even more visible in Figure 4D, which summarizes the average area obtained for the samples containing 2, 8, and 20 µg/mL of pyriproxyfen.

### 2.6. System Life Use

As yet, the life use of graphene-based columns is uncertain. In this study, we hope to bring some light to this aspect. The analyses were performed by injecting the sample without any preparation (despite adding 25% acetonitrile to keep the analytes soluble). After its hundredth analysis, the SiGO column (extraction column) presented for the first time a carry-over effect (Figure 5). Additionally, the column showed a blockage after about the 120th analysis. The column was washed in the back flush mode, removing the blockage, but the carry-over effect remained.

Additionally, after the washing, the column lost its repeatability compared to the previous separations. Therefore, the column was not considered helpful after this for this application. In contrast, although the SiGO-C18ec column (analytical) has an unknown life-use time, it has been used in uncountable separations (at least more than 300) and remains usable. It is essential to highlight that the SiGO-C18ec column used in this study has been used only as the analytical column, which may favor its durability.

### 2.7. Peak Shape Comparison with C18 Analytical Column

Another variable explored was the effect of the analytical column in the chromatograms obtained by the described column-switching method. The SiGO-C18ec (0.3 × 100 mm, 10 µm FPPs) column was compared to a C18 column (50 × 2.6 mm, 1.9 µm SPPs) using the same chromatographic method (Figure 6). For the abamectin B1a, it is clear that the C18 column resulted in thinner peaks than the SiGO-C18ec column for all the concentrations explored. No tail was observed for this compound in either column. However, a similar tailing was observed in both columns for the pyriproxyfen analysis, indicating that it may be caused in the SiGO extraction column. Although both columns presented tails for the pyriproxyfen peak, the tailing was higher (proportionally to the peak height) on the SiGO-C18ec column. In some separations using the SiGO-C18ec column, especially at concentrations ≥ 15 µg/mL, additional analysis was necessary to understand better and define when the tail ends and the baseline starts. One relevant highlight that must be taken into account here is the difference between the particle diameter and its porosity between the columns compared. The SiGO-C18ec is packed with 10 µm FPPs, while the C18 column is packed with 1.9 µm SPPs. Usually, reducing the particle diameter results in better performance, and reducing the particle core porosity does the same [46,47]. Additionally, as already mentioned, the SiGO-C18ec columns are in an early, underdevelopment stage, and their comparison to the well-established C18 columns needs to be conducted carefully. A positive affirmation about the SiGO-C18ec columns is that it is possible to confirm that it can be applied as a stationary phase and replace C18 columns occasionally. As a negative statement about the SiGO-C18ec columns, it is possible to affirm that it still demands several developments to be at a comparable stage with C18.

The conclusions drawn via visual observation of peak shape were corroborated by analyzing the peak capacity performance parameter. In the proposed chromatographic method, the SiGO-C18ec column exhibited a peak capacity of six peaks in 15 min. In comparison, the C18 column demonstrated a peak capacity of 12 peaks under similar conditions, both at a concentration of 8 µg/mL. The peak capacity was determined at 13.4% of the peak height (equivalent to 4ơ). However, the peak capacity observed in this study was lower than that reported in the literature for columns containing the same phase (SiGO-C18ec particles of 10 µm diameter). Notably, for hormone separation, a peak capacity of around 27 peaks in 15 min was reported [41], while for the separation of multiclass analytes, the reported peak capacity was 14 peaks in 15 min [42]. A comparison with other graphene-based stationary phases revealed similar outcomes. For instance, a peak capacity of approximately 19 peaks in 8 min was obtained to separate aromatic compounds using a stationary phase composed of octadecylsilane-functionalized graphene oxide coated over silica particles [37]. In another study, separations of anilines utilizing graphene oxide coated over silica particles displayed an estimated peak capacity of 15 peaks in 12 min [36]. Additionally, a significantly higher result, estimating a peak capacity of 57 peaks in 18 min, was reported for separating alkylbenzenes using a stationary phase composed of octadecyl amine and serine-derived carbon dot-modified silica gel [48]. The estimated peak capacity reported in the above three instances was evaluated based on chromatograms available in the references.

## 3. Material and Methods

### 3.1. Reagents and Standards

LC-grade acetone, acetonitrile, and MS-grade high-purity (>98%) formic acid were purchased from VWR International (Fontenay-sous-Bois, France). Deionized ultrapure water (H_2_O) was produced in the lab by a Milli-Q^®^ Millipore system (Burlington, NJ, USA). The SiGO and SiGO-C18ec fully porous particles (FPPs) were produced as described by Maciel et al. [41]. The C18 column packed with 1.9 µm superficially porous particles (SPPs) was a Hypersil GOLD 50 × 2.1 mm (Thermo Electronic Corporation, San Jose, CA, USA). Pyriproxyfen and abamectin B1a were acquired from Cymit Chimica (Barcelona, Spain). The orange-flavored carbonated soft drinks comprising 8% orange juice (Kas Naranja, PepsiCo, Spain) and 1.7% (Schweppes Naranja, Schweppes, S.A. Spain) were both acquired in a local market.

### 3.2. Sample Preparation

The analytes’ standards were solubilized to a 1000 µg/mL solution in pure acetone and then diluted to 100 µg/mL in 90% acetonitrile and 10% acetone solution. The 100 µg/mL pesticide solution produced the 0.5–25 µg/mL samples by spiking the analytes in the matrix. The samples comprised 25% organic fraction and 75% orange-flavored carbonated soft drinks. The soft drink with 8% orange juice in the composition spiked with the analytes was used for the calibration curves, accuracy and precision, and comparison with C18 column experiments. Solutions of 2, 8, and 20 µg/mL of each orange-flavored carbonated soft drink and water (25% organic and 75% water) were used to evaluate the matrix effect. The orange-flavored carbonated soft drinks were selected as the matrix because, although it is a complex matrix, it can be directly injected into the system without pretreatment.

### 3.3. Analytical Instrumentation

A liquid chromatography Accela LC system composed of an auto-sampler, a column temperature controller, and a quaternary pump (Thermo Electronic Corporation, San Jose, CA, USA) coupled with a TSQ Quantum Access triple-quadrupole mass spectrometer (MS) fitted with an electrospray ion (ESI) source (Thermo Electronic Corporation, San Jose, CA, USA) was used in this work. A 100 µL loop was placed in the injection valve. Analyses were carried out using lab-made SiGO and SiGO-C18ec columns prepared as described by Borsatto et al. [42], using 100 × 0.3 mm tubing and 5 µm SiGO FPPs and 10 µm SiGO-C18ec FPPs. The column-switching system was assembled as shown in Figure 7. The SiGO column (extraction column) was connected between the injection valve and the MS valve, and the SiGO-C18ec column was connected between the MS valve and the ESI inlet.

### 3.4. Analytical Methods

The mobile phase consisted of H_2_O (A) and acetonitrile (B), both containing 0.1% of formic acid as an additive to enhance the positive electrospray (ESI) ionization process. In addition, a gradient method was used, as described in Table 2.

An equilibration gradient drove the mobile phase composition back to 100% H_2_O in 5 min, followed by an additional 10 min of constant 100% H_2_O. The flow rate was kept constant at 50 µL/min, except for the loading flow rate experiment, in which 25 and 100 µL/min flows were also explored. The MS parameters were spray voltage of 4000 V, N_2_ desolvation gas flow of 20 L/h, and capillary temperature of 400 °C. A multiple reaction monitoring (MRM) mode was used to observe the analytes. The quantifying transitions were 322 > 185 (collision energy of 20 V) for pyriproxyfen and 895 > 449 (collision energy of 40 V) for abamectin B1a. The qualifying transitions were 322 > 96 (collision energy of 40 V) for pyriproxyfen and 895 > 305 (collision energy of 70 V) for abamectin B1a.

## 4. Conclusions

This study investigated the applicability and limitations of a column-switching system composed exclusively of columns packed with graphene-based stationary phases. The samples of orange-flavored soft drinks with spiked analytes (25% of acetonitrile added to the samples) were directly injected without pretreatment or sample preparation. It was observed that the proposed column-switching system presented good linearity in the calibration curve and reasonable accuracy and precision (% RSD) for pyriproxyfen analysis, but not for abamectin B1a analysis. Furthermore, it demonstrated that this system could be applied to reliable quantitative analysis, but it is not multipurpose and presents limitations. The main restrictive factor was the limited capability of the SiGO extraction column to retain some analytes well. It was also observed that the proportion of orange juice in the matrix composition affected the results.

Additionally, it was noted that the life use of the SiGO extraction column was about 120 injections in the explored conditions, blockages and carry-over effects being observed. In the last stage of this study, the analytical SiGO-C18ec column was compared with a commercial C18 column. It was noted that although the C18 column presented better performance with thinner peaks and less tail, owing to the much smaller particle size of the stationary phase, the SiGO-C18ec columns can be applied for the same use with no practical restriction. In short, graphene-based particles are promising stationary phases for liquid chromatography and can be applied in both dimensions of a column-switching system. Although columns packed with these materials have been finding current applications in LC, improvements are still needed to increase the range of applications and life use. If these phases keep being developed and optimized, soon it might become an additional solution for separation science applications. Optimizations in the injection process and the detection method could improve the sensibility and applicability of this system proposed and might be an interesting topic to be investigated in the future. Furthermore, ramifications of this work to investigate different analytes in the various matrix could lead to a better understanding of these phases and help to draw the boundary conditions for these systems.

## Figures and Tables

**Figure 1 molecules-28-04999-f001:**
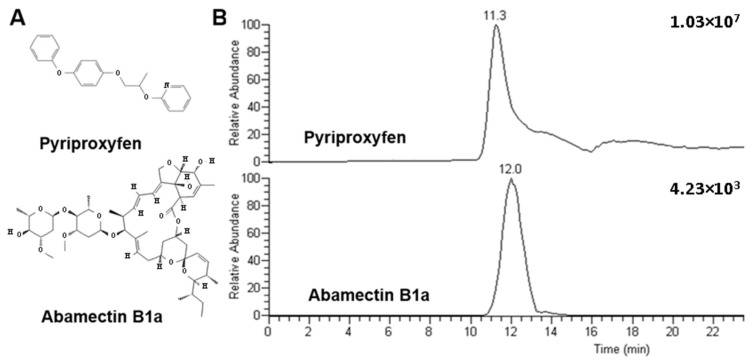
Structure of the analytes evaluated (**A**) and chromatograms obtained in the column-switching method described (**B**). Separation conditions: loading time of 3 min at 50 µL/min, linear gradient from 50% to 100% B in 15 min followed by an additional 5 min of 100% of B; 100 µL of the sample at a concentration of 10 µg/mL of each analyte was injected.

**Figure 2 molecules-28-04999-f002:**
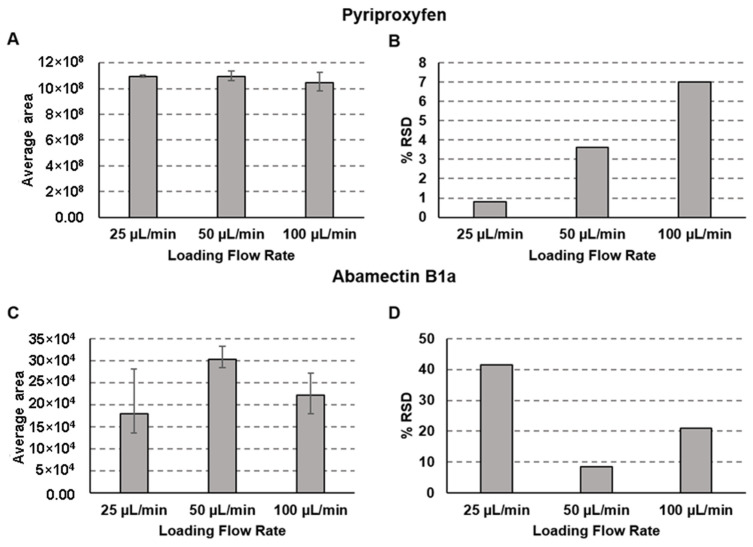
The area obtained with loading flow rates of 25, 50, and 100 µL/min (**A**) and % RSD for each loading flow rate (**B**) for pyriproxyfen and the area obtained with loading flow rates of 25, 50, and 100 µL/min (**C**) and % RSD for each loading flow rate (**D**) for abamectin B1a. Separation conditions: loading time of 3 min at 50 µL/min, linear gradient from 50% to 100% B in 15 min followed by an additional 5 min of 100% of B; 100 µL of the sample at a concentration of 10 µg/mL of each analyte was injected. The analyses were performed in triplicate for each flow rate, and the error bars represent the maximal and minimal values observed experimentally.

**Figure 3 molecules-28-04999-f003:**
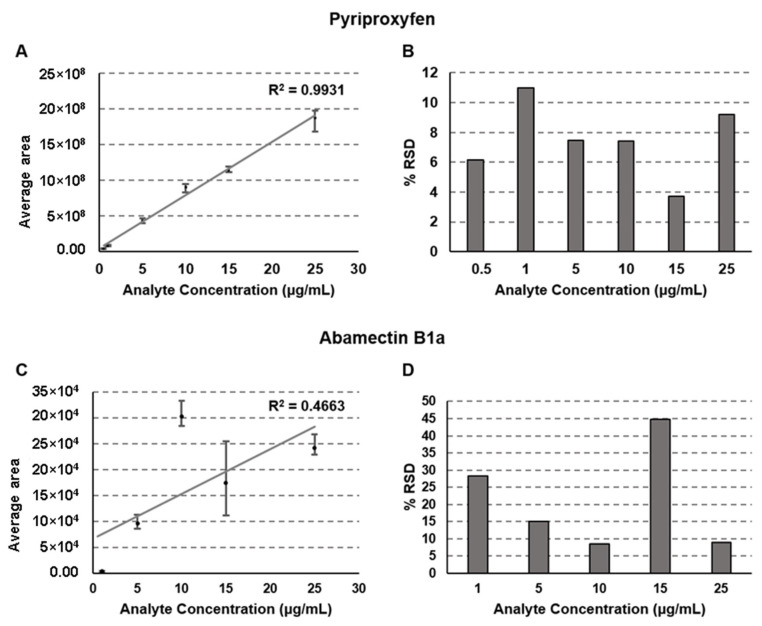
The calibration curve (**A**) and % RSD for each point of the calibration curve (**B**) for pyriproxyfen and the calibration curve (**C**) and % RSD for each point of the calibration curve (**D**) for abamectin B1a. Separation conditions: loading time of 3 min at 50 µL/min, linear gradient from 50% to 100% B in 15 min followed by an additional 5 min of 100% of B; 100 µL of the sample was injected. The analyses were performed in triplicate for each concentration and the error bars represent the maximal and minimal values observed experimentally.

**Figure 4 molecules-28-04999-f004:**
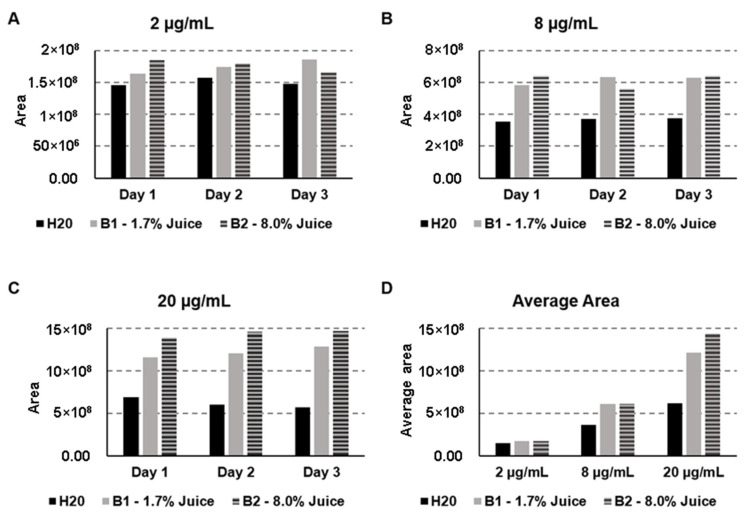
Matrix effect in the obtained area of pyriproxyfen for the 2 µg/mL (**A**), 8 µg/mL (**B**), 20 µg/mL (**C**), and a comparison of the average for the three concentrations (**D**). Separation conditions: loading time of 3 min at 50 µL/min, linear gradient from 50% to 100% B in 15 min followed by an additional 5 min of 100% of B; 100 µL of analytes were injected.

**Figure 5 molecules-28-04999-f005:**
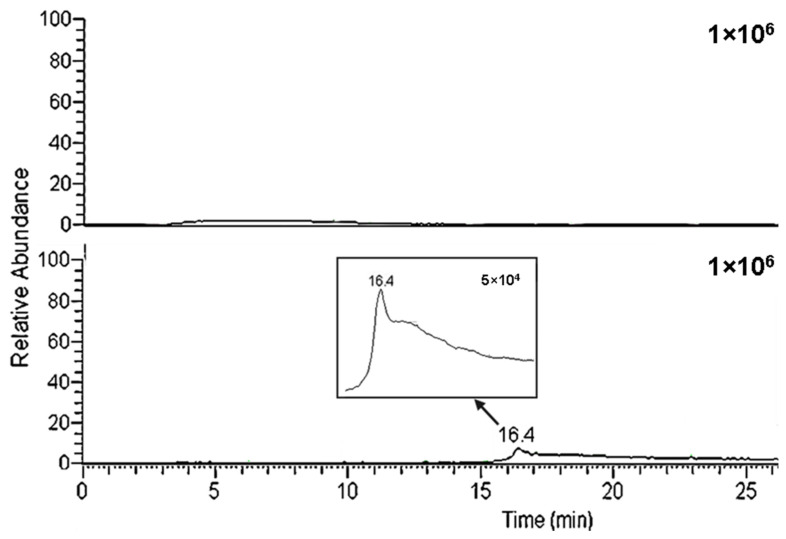
Blanks were obtained after 50 and 100 analyses performed in the same column. Separation conditions: loading time of 3 min at 50 µL/min, linear gradient from 50% to 100% B in 15 min followed by an additional 5 min of 100% B. No sample was injected.

**Figure 6 molecules-28-04999-f006:**
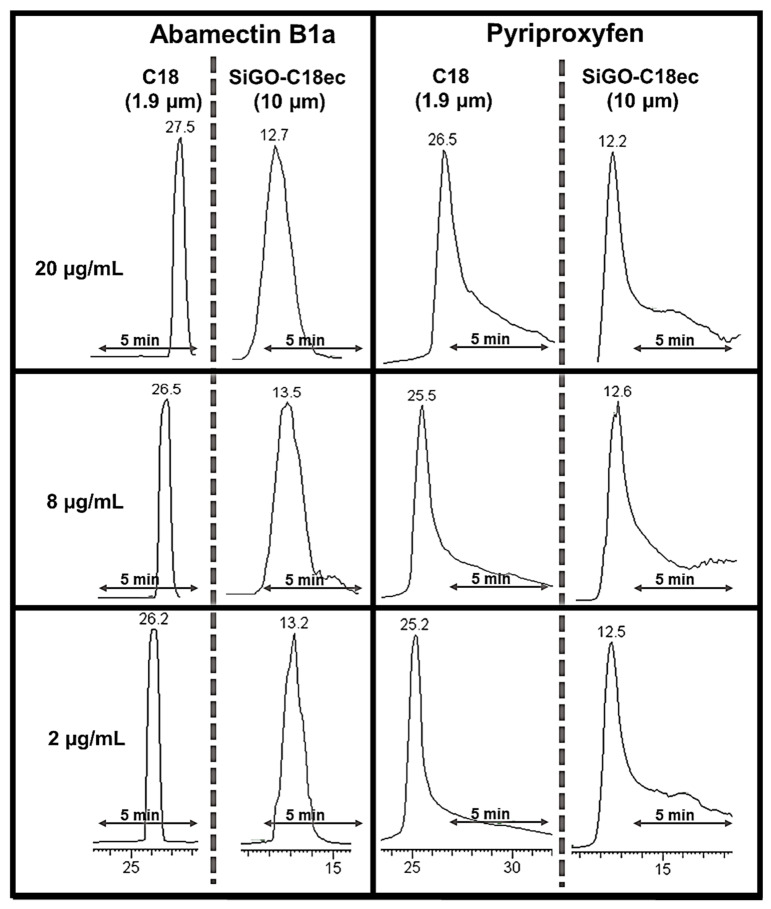
Comparisons of peak shape obtained for abamectin B1a and pyriproxyfen using C18 and SiGO-C18ec column as an analytical column in the column-switching system. Separation conditions: loading time of 3 min at 50 µL/min, linear gradient from 50% to 100% B in 15 min followed by an additional 5 min of 100% of B; 100 µL of analytes were injected.

**Figure 7 molecules-28-04999-f007:**
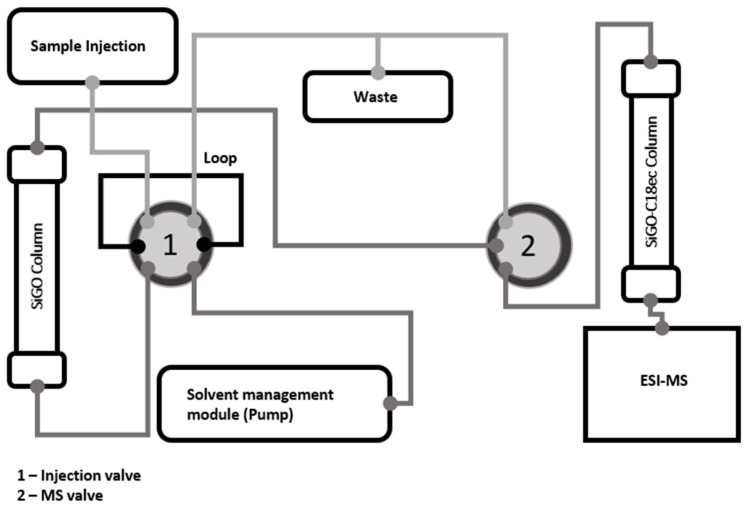
Scheme illustrating the column-switching system evaluated.

**Table 1 molecules-28-04999-t001:** Intra-day and inter-day accuracy and precision (% RSD) obtained using the described column-switching system for both analytes. Separation conditions: loading time of 3 min at 50 µL/min, linear gradient from 50% to 100% B in 15 min followed by an additional 5 min of 100% of B; 100 µL of sample was injected.

Compound		Pyriproxyfen			Abamectin B1a		
Concentration (µg/mL)		2	8	20	2	8	20
Expected Area *		190,000,000	610,000,000	1,450,000,000	84,362	136,104	239,587
Day 1	Area	176,290,988	612,883,132	1,438,632,466	4899	120,972	274,393
	Intra-Day RSD %	5.8	7.8	3.1	172.8	22.6	9
	% Accuracy **	92.8	100.5	99.2	5.8	88.9	114.5
Day 2	Area	156,108,411	621,257,647	1,501,904,177	7836	99,444	378,336
	Intra-Day RSD %	8.9	13.3	3.3	96.1	31.3	3
	% Accuracy **	82.2	101.8	103.6	9.3	73.1	157.9
Day 3	Area	211,749,192	567,532,037	1,399,813,509	12,072	282,730	372,730
	Intra-Day RSD %	3.2	5.3	5.4	344.6	18.9	17
	% Accuracy **	111.4	93.0	96.5	14.3	207.7	155.6
Average Inter-Day Area		181,382,863	600,557,605	1,446,783,384	8269	167,715	341,820
Inter-Day RSD % ***		12.7	3.9	2.9	35.6	48.8	14
Inter-Day % Accuracy **		95.5	98.5	99.8	9.8	123.2	142.7

* Calculated based on the calibration curve. ** Accuracy is calculated based on the expected area from the calibration curve. *** Calculated based on the intra-day averages.

**Table 2 molecules-28-04999-t002:** HPLC column-switching events.

Event	Time	% B	Injection Valve Position	MS Valve Position
Pre-run	0 min	0%	Loading	Waste
Loading	0 to 3 min	0%	Inject	Waste
Gradient	3 to 18 min	50% to 100%	Inject	ESI-MS
Isocratic	18 to 23 min	100%	Inject	ESI-MS

## Data Availability

Data will be made available on request.

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
