# Peer review of "Applicability and Limitations of a Capillary-LC Column-Switching System Using Hybrid Graphene-Based Stationary Phases"

_molecules, 2023, doi:10.3390/molecules28134999_

Round 1

Reviewer 1 Report

The manuscript is well written, but some revisions are needed to improve its quality before being published.

My comments and suggestion for revision are as follows:

# In the Abstract section:

mention the results for linearity for both analytes, including the concentration range of the calibration curve.

mention the results for matrix effects

Based on the progress made in the current study, mention further research needed for improvement in the future 

# In the Introduction section:

Provide brief introduction to the two analytes studied

Provide the reasons why were they selected? for example, what do these two analytes offer for the development of the analytical technique being studied?

# In the Methodology and Results section:

Why LOD and/or LOQ were not estimated?  

Was there any effect from detector stability to the results of analysis, especially for Abamectin measurement? Since an internal standard was not used in this study, the effect of detector stability on the measurement was likely not observed.

Author Response

Molecules-2422074 – authors’ response

Reviewer # 1

The manuscript is well written, but some revisions are needed to improve its quality before being published.

Response:

Dear reviewer

We appreciate your comments and questions, and we are thankful for the time that you have dedicated to evaluating this manuscript. They were helpful and assisted in improving the manuscript quality. We hope to have addressed your suggestions properly. Please, find below the response, topic by topic of each suggestion, comment, or question.

My comments and suggestion for revision are as follows:

# In the Abstract section:

Response:

We are glad for your comments and suggestions. Unfortunately, due to the limitation of the number of words (200 words) in the abstract,  we could not add all the information in the abstract, but we made an effort to satisfy your suggestion and fit the solicited information in the 200 words of the abstract.

mention the results for linearity for both analytes, including the concentration range of the calibration curve.

Response:

Thank you for your suggestion. To satisfy this comment we added the information underlined below in the abstract.

“The proposed system could be successfully applied to the pyriproxyfen analysis in a concentration range between 0.5 to 25 µg/mL presenting a linearity of R2 = 0.9931 and an intra-day and inter-day accuracy between 82.2⸺111.4% (RSD < 13.3%) and 95.5%⸺99.8% (RSD < 12.7%), respectively.”

mention the results for matrix effects

Response:

We are thankful for this suggestion. To satisfy this comment we added the information underlined below in the abstract

“Furthermore, the matrix composition affected the area observed for the pyriproxyfen, been the higher the concentration of orange juice in the soft-drink, the higher pyriproxyfen the signal observed.”

Based on the progress made in the current study, mention further research needed for improvement in the future

Response:

We appreciated this suggestion and intended to add it to the abstract, but, unfortunately, due to the word number limitation in the abstract, it was not possible. Once we consider your suggestion relevant to the improvement of the paper, we decided to address this suggestion in the conclusion section. The sentence below was added to the text.

“Optimizations in the injection process and the detection method could improve the sensibility and applicability of this system proposed and might be an interesting topic to be investigated in the future. Furthermore, ramifications of this work to investigate different analytes in diverse matrix could leady do a better understanding of these phases and help to draw the boundary conditions for this systems.”

# In the Introduction section:

Provide brief introduction to the two analytes studied

Response:

We are thankful for this comment. We added the information in the last paragraph of the introduction below.

“Pyriproxyfen and abamectin B1a are insecticides employed to control several pest’s species in diverse cultures. They have been selected as model analytes for this study because quantitative analysis has already been reported in the literature for both compounds in orange and orange by-products samples [44,45].

Provide the reasons why were they selected? for example, what do these two analytes offer for the development of the analytical technique being studied?

Response:  

We are thankful for this comment, and we agree on the importance of this statement. Usually, if this paper were submitted to other journals, we would, for sure, address this suggestion. But, for the molecules style, the results section is placed just after the introduction, and the results section presents the sub-section 2.1 “Selection of the analytes".  So, due to that, it was decided to let this information only in the results section not avoid duplication of information in sections to close to each other.

# In the Methodology and Results section:

Why LOD and/or LOQ were not estimated? 

Response:

We are thankful for your observation

Initially, it was expected to evaluate the LOD and LOQ parameters, but it was not possible due to the blocking of the extraction column (SiGO column). A blockage in the extraction column was already expected before the beginning of the analysis, so the experimental series was designed to collect as much as information before the blocking happened. The experimental series was planned to be executed in the following order: (i) the loading method, (ii) the calibration curve and inter-day and intra-day repeatability, (iii) comparison to the C18 column, (iv) the matrix effect, (v) the LOD and LOQ experiments.  Unfortunately, the SiGO column presented a blockage during the matrix effect experiment. A fourth matrix, with 7% orange juice in the composition of the soft drink was planned to be evaluated, no data could be obtained because of the column blocking.  Because of this limitation, the experimental series was finished in this step.

We would like to highlight that, to the best of our knowledge, this is the first columns-witching system reported composed exclusively with graphene-based packed columns, so limitations are expected due to the early-stage development of this analytical strategy. Although not all data expected could be obtained, we considered that the results observed, including the limitations described, were significant enough to be published and could help or inspire other researchers that work with similar topics to find out solutions for their problems or to improve the system proposed.

Was there any effect from detector stability on the results of analysis, especially for Abamectin measurement? Since an internal standard was not used in this study, the effect of detector stability on the measurement was likely not observed.

Response: 

We are thankful for your comment and observation. No study was performed to evaluate the effect of the detector in the analysis. The MS method was optimized by analyzing the pure standards by direct infusion, but no further details were explored.  Probably, an optimization of the detection method could improve the results. Since our focus was to investigate the applicability and limitations of the “chromatographic parameters” of the proposed column-switching system, we limited the variable to be evaluated and optimized in this work focusing only on the liquid chromatography system. Addressing your suggestion made before, we added the information underlined below in the conclusion, pointing out topics to be investigated and optimized in future work.

“Although columns packed with these materials have been finding current applications in LC, it still demands improvements to increase the range of applications and life-use. If these phases keep being developed and optimized, shortly, it might become an additional solution for separation science applications. Optimizations in the injection process and the detection method could improve the sensibility and applicability of this system pro-posed and might be an interesting topic to be investigated in the future. Furthermore, ramifications of this work to investigate different analytes in diverse matrix could leady do a better understanding of these phases and help to draw the boundary conditions for this systems.”

Reviewer 2 Report

The reviewed paper is devoted to the application of two types of graphene modified silica materials for the preconcentration and HPLC separation of only two insecticides (pyriproxyfen and abamectin B1a). The novelty of the work is rather moderate as both graphene containing phases have been described by the authors in many publications as well as their applications in SPE, column switching concentrators and stationary phases in HPLC. Also, these phases has been also used for the analysis of herbicides in sugar cane (J. Chromatogr. A 1687(2023) 463690). The chromatographic performance of graphene coated columns is also not too impressive with severe peak broadening and low column efficiency. As result a relatively poor accuracy 12.7 -13.3 is reported for the analytical method. In conclusion, the most of the obtained results show the disadvantage of the developed stationary phase (except of sample pretreatment), so I am afraid that this paper cannot be recommended for the publication in the present form in Molecules. Probably, the authors should focus only on the use of graphene coated silica only for the preconcentration. Anyway, there are few comments.

Comments

1.      It is strongly recommended to specify the novelty of the investigation.

2.      The column efficiency is reported for the prepared chromatographic columns. Please, specify column efficiency and compare with analogues (at least with other graphene coated silicas).

3.      No baseline separation was achieved for this pair of analytes, which raises a question about necessity of using chromatographic separation step before triple quad mass spectrometer detection. Otherwise, the separation conditions should be optimized.

4.      Fig 1. Due to poor separation efficiency no baseline separation of chromatographic peaks is obtained for pyriproxyfen and abamectin B1a. What is the reason to use graphene coated separation column in this case instead of common, more efficient, octadecylsilica columns?

5.      Fig 3, D. Very high RSD values of 45% and 28% are reported for abamectin B1a concentrations at 15 and 1 mkg/mL, respectively. This resulted in unacceptable correlation coefficients (R2 = 0.4663) calculated for this insecticide.

6.      It would be also useful to use UV detection in combination to check baseline profile after sample injection to identify a possible reason for carry-over effects.

Author Response

Molecules-2422074 – authors’ response

Reviewer # 2

The reviewed paper is devoted to the application of two types of graphene modified silica materials for the preconcentration and HPLC separation of only two insecticides (pyriproxyfen and abamectin B1a). The novelty of the work is rather moderate as both graphene containing phases have been described by the authors in many publications as well as their applications in SPE, column switching concentrators and stationary phases in HPLC. Also, these phases have been also used for the analysis of herbicides in sugar cane (J. Chromatogr. A 1687(2023) 463690). The chromatographic performance of graphene coated columns is also not too impressive with severe peak broadening and low column efficiency. As result a relatively poor accuracy 12.7 -13.3 is reported for the analytical method. In conclusion, most of the obtained results show the disadvantage of the developed stationary phase (except of sample pretreatment), so I am afraid that this paper cannot be recommended for the publication in the present form in Molecules. Probably, the authors should focus only on the use of graphene coated silica only for the preconcentration. Anyway, there are few comments.

Response:

Dear reviewer

We appreciate the time you have dedicated to evaluating this manuscript and we are thankful for your observations, comments, and suggestions. We consider that your contribution was important for the improvement of the work. We understand your point of view and consider it correct. We would like, kindly, to present our point of view. Graphene-based columns are in the early-development stage, so limitations are expected and are being overcome, slowly, by set-by-step as further experiments are performed. As you mentioned, we have been working on trying to better understand the graphene-based phases applied to liquid chromatography, including its advantages and, also, disadvantages. To the best of our knowledge, it is the first reported application of a system composed exclusively of graphene-based columns. Though a conventional system, composed of commercially available columns, will present better performance, our target was to investigate and present the applicability and limitations of a column-switching system composed exclusively of graphene-based columns. We understand that bad results (limitations) are also important for science because they can help other researchers to avoid mistakes or inspire then-selves and improve them. We hope to have addressed your suggestions, comments, and questions satisfactorily.

Comments

  1. It is strongly recommended to specify the novelty of the investigation.

Response:

Thank you for your observation. We added the sentence underlined below in other to address your suggestion and make our objective clear.

“This work evaluates the applicability and limitations of a column-switching system consisting exclusively of columns packed with graphene-based stationary phases. To the best of our knowledge, no work describes the use of column-switching system composed exclusively of graphene-based packed columns to any application, been this a vacancy to be investigated.”

  1. The column efficiency is reported for the prepared chromatographic columns. Please, specify column efficiency and compare with analogues (at least with other graphene coated silicas).

Response:

We are glad for your comment. Though we agree it could be interesting to add performance evaluation to this paper we decided to not do it once similar works have been published in the literature concerning this topic. The performance parameter of the graphene-based analytical column has already been explored for other analytes in other publications, mentioned in the introduction of this manuscript in references number 42 and 43.

  1. Maciel, E.V.S.; Borsatto, J.V.B.; Mejia-Carmona, K.; Lanças, F.M. Application of an In-House Packed Octadecylsilica-Functionalized Graphene Oxide Column for Capillary Liquid Chromatography Analysis of Hormones in Urine Samples. Anal Chim Acta 2023, 1239, doi:10.1016/j.aca.2022.340718.
  2. Borsatto, J.V.B.; Maciel, E.V.S.; Lanças, F.M. Investigation of the Applicability of Silica-Graphene Hybrid Materials as Stationary Phases for Capillary Liquid Chromatography. J Chromatogr A 2022, 1685, 463618, doi:10.1016/j.chroma.2022.463618
  3. No baseline separation was achieved for this pair of analytes, which raises a question about necessity of using chromatographic separation step before triple quad mass spectrometer detection. Otherwise, the separation conditions should be optimized.

Response:

We are glad for your observation and question. The similar retention time of the analytes was purposely selected by us for this work. Your comment was important because helped us to identify that the reasons for the selection of these analytes were not clear enough in the text, so we decided to add extra information, underlined below.

“Despite these structural differences, pyriproxyfen and abamectin B1a presented closed retention times in the column-switching system composed of the columns packed with SiGO and SiGO-C18ec (Figure 1-B). The similarity in the retention time indicated that both analytes were similarly retained in the proposed column-switching system. This characteristic was important to evaluate if the proposed column-switching system, composed exclusively of columns packed with graphene-based phases, is multipurpose or not. If the system demonstrated successful quantitative analysis for only one of the analytes, it would suggest its limitations as a multipurpose system. Conversely, if it provided suitable quantitative analysis for both compounds, further investigation would be necessary to confirm its multipurpose capabilities of the system.”

  1. Fig 1. Due to poor separation efficiency no baseline separation of chromatographic peaks is obtained for pyriproxyfen and abamectin B1a. What is the reason to use graphene coated separation column in this case instead of common, more efficient, octadecylsilica columns?

Response:

We are thankful for your observation. This statement is very welcomed and would be true if the objective of this study was to present a new analytical method to replace the current applicable LC methods for pesticide analysis. But it was not our intention. We intend to present a novel system composed exclusively of graphene-based columns and investigate the positive points and also the negative points of this system. This is a new system, and it is expected to present reduced performance in comparison to a better-established system. With this work, we intend to present a new concept that if well-optimized might become an additional solution for analytical analysis.  C18, as an example, has been submitted for decades of improvements until they present excellent performance today while SiGO and SiGO-C18ec phases are new and are being still optimized.

  1. Fig 3, D. Very high RSD values of 45% and 28% are reported for abamectin B1a concentrations at 15 and 1 mkg/mL, respectively. This resulted in unacceptable correlation coefficients (R2 = 0.4663) calculated for this insecticide.

Response:

We are thankful for your comments. This shows that the proposed system is not applicable to abamectin B1a analysis. Though this result is “negative” it also indicates that the proposed system is not multipurpose. It is a bad result, but it is a result that point out an important characteristic of this system and need to be reported. It may help future research to avoid a mistake or, maybe, see some think we did not and improve the system in future work.

  1. It would be also useful to use UV detection in combination to check baseline profile after sample injection to identify a possible reason for carry-over effects.

Response:

We are glad for this suggestion. We would like to address it and introduce this in the paper, unfortunately, it not possible to obtain the data additional data because the column used is blocked (as reported). In future work, it can be evaluated. To attempt to satisfy this suggestion we added the phrases below in the conclusion suggesting improvements to be done in future work.

“Optimizations in the injection process and the detection method could improve the sensibility and applicability of this system proposed and might be an interesting topic to be investigated in the future. Furthermore, ramifications of this work to investigate different analytes in diverse matrix could leady do a better understanding of these phases and help to draw the boundary conditions for this systems.”

Reviewer 3 Report

Dear Authors,

Congrats on your manuscript. Here are some suggestions for your consideration to make it better

In abstract, % in front of RSD could be removed?

Also, "life-use of about 120" consider adding in what was the minimum life?

For the abstract picture, do add in simple legend for the graph.

Do consider adding to let readers know why your team has chosen pyriproxyfen and abamectin as the analytes and orange-flavored carbonated soft drinks as a matrix.

Is there a problem that this work seeks to address/solve? Is this the first time graphene-based columns are used in this way?

Do elaborate more on the "matrix effect".

Author Response

Molecules-2422074 – authors’ response

Reviewer # 3

Dear Authors,

Congrats on your manuscript. Here are some suggestions for your consideration to make it better

 Response:

Dear reviewer,

We are glad for your kind comments. Thank you for taking the time to review our manuscript and for your feedback. Your contributions are highly valued, and we hope to have addressed them adequately. Please, find below the response, topic by topic of each comment, suggestion, or question.

In abstract, % in front of RSD could be removed?

Response:

Thank you for your suggestion. It was removed.

Also, "life-use of about 120" consider adding in what was the minimum life?

Response:

We are thankful for this observation. The word "about" was removed to indicate that the column could be used at least for 120 injections. The abstract was modified below:

“Additionally, the SiGO extraction column presented a life-use of 120 injections for this matrix.”

For the abstract picture, do add in simple legend for the graph.

Response:

thank you for your suggestion. The caption was added as below.

“Graphical abstract caption. Illustration representing the proposed column-switching system using exclusively graphene-based columns in the analysis of pesticides in orange flavored soft drinks.”

Do consider adding to let readers know why your team has chosen pyriproxyfen and abamectin as the analytes and orange-flavored carbonated soft drinks as a matrix.

Response:

We are thankful for this comment. We added the information in the last paragraph of the introduction below.

“Orange-soft drinks are interesting matrix to evaluate this proposed column-switching system. Thought this matrix is complex and presents solid material in suspension, it also can be directly injected in the column-switching system without pre-treatment. Pyriproxyfen and abamectin B1a are insecticides employed to control several pest’s species in diverse cultures. They have been selected as model analytes for this study because quantitative analysis has already been reported in the literature for both compounds in orange and orange by-products samples [44,45]”

  1. Ferrer, C.; Martínez-Bueno, M.J.; Lozano, A.; Fernández-Alba, A.R. Pesticide Residue Analysis of Fruit Juices by LC–MS/MS Direct Injection. One Year Pilot Survey. Talanta 2011, 83, 1552–1561, doi:10.1016/j.talanta.2010.11.061.
  2. by Liquid Chromatography–Electrospray Tandem Mass Spectrometry. J Chromatogr A 2003, 992, 133–140, doi:10.1016/S0021-9673(03)00325-X.

Is there a problem that this work seeks to address/solve? Is this the first time graphene-based columns are used in this way?

Response:

Thank you for this observation. Our idea was to investigate a column-switching system composed exclusively of graphene-based columns. Graphene-based columns were already used in the extraction and analytical columns in this kind of system, but to the best of our knowledge, there is no report about the simultaneous use of these kinds of columns in column switching systems. To clarify this information to the reader, we added the underlined phrase below in the introduction.

“This work evaluates the applicability and limitations of a column-switching system consisting exclusively of columns packed with graphene-based stationary phases. To the best of our knowledge, no work describes the use of column-switching system composed exclusively of graphene-based packed columns to any application, been this a vacancy to be investigated.”

Do elaborate more on the "matrix effect".

Response:

Thank you for your suggestion. We added a short statement explaining the expression "matrix effect" at the beginning of the sub-section "2.5. Matrix effects” as below

“The matrix composition might affect the analytical signal (peak area, as an example) observed, interfering with the analysis; this effect is called "matrix effect”.

Reviewer 4 Report

The manuscript entitled Applicability and limitations of a capillary-LC column-switching system using hybrid graphene-based stationary phases focused on the possibility of developing an online LC-MS method in food detection. To realize the destination, the authors chose graphene oxide, one nano material, as both an extraction and separation media for two chemicals, pyriproxyfen and abamectin B1a. After the validation of their extraction and separation strategy, the authors carefully test the use number of their self-made column and made a comparison between the self-made column and the C18 column. The manuscript was written well. It is remarkable that the authors not only give the advantages of the strategy they have developed, but also pointed out that this extraction and separation strategy needs to be further enhanced to have a wider general use. This work is a good attempt to apply graphene to the field of online extraction and separation analysis. Therefore, this manuscript is recommended to be accepted by the journal Molecules. Here are some errors and suggestions for the authors:

1)     In the legend of Figure 1, only (A) appeared, the authors should add (B).

2)     In the legend of Figure 2, it seems that two spaces were found in the second line. And one space should be deleted.

3)     In Figure 1, each chemical structure of Figure 1A seems too small and Figure1B seems too wide. Therefore, Figure 1B should be adjusted to be a little narrow and Figure1A should be changed wider.

4)     In Figure 2, as the context of the manuscript has said that the column can only be used for about 120 injections, but the legend of Figure 2 did not mention whether the authors used the same column or not. Therefore, the authors should add some word to explain it.

5)     Error bars appeared in both Figure 2 and Figure 3. However, the authors did not mention the replication number of relative experiments. The authors should explain it in the legend.

Author Response

Molecules-2422074 – authors’ response

Reviewer # 4

The manuscript entitled Applicability and limitations of a capillary-LC column-switching system using hybrid graphene-based stationary phases focused on the possibility of developing an online LC-MS method in food detection. To realize the destination, the authors chose graphene oxide, one nano material, as both an extraction and separation media for two chemicals, pyriproxyfen and abamectin B1a. After the validation of their extraction and separation strategy, the authors carefully test the use number of their self-made column and made a comparison between the self-made column and the C18 column. The manuscript was written well. It is remarkable that the authors not only give the advantages of the strategy they have developed, but also pointed out that this extraction and separation strategy needs to be further enhanced to have a wider general use. This work is a good attempt to apply graphene to the field of online extraction and separation analysis. Therefore, this manuscript is recommended to be accepted by the journal Molecules. Here are some errors and suggestions for the authors:

Response:

Dear reviewer,

We appreciated the time you have dedicated to evaluating this work and we welcome your comments, corrections, and suggestions. We hope to address your suggestions properly. Please, find below the response, topic by topic of each comment, suggestion, or question.

1)     In the legend of Figure 1, only (A) appeared, the authors should add (B).

Response:

Thank you for your correction. “B” was added as below.

“Figure 1. Structure of the analytes evaluated (A) and Chromatograms obtained in the column-switching method described (B). Separation conditions: Loading time of 3 min at 50 µL/min, linear gradient from 50 to 100% B in 15 min followed by an additional 5 min of 100% of B. 100 µL of the sample at a concentration of 10 µg/mL of each analyte were injected.”

2)     In the legend of Figure 2, it seems that two spaces were found in the second line. And one space should be deleted.

Response:

We appreciate your correction. The figure caption was modified as below.

“Figure 2. The area obtained with loading flow rates of 25, 50, and 100 µL/min (A) and % RSD for each loading flow rate (B) for pyriproxyfen and the area obtained with loading flow rates of 25, 50, and 100 µL/min (C) and % RSD for each loading flow rate (D) for abamectin B1a. Separation conditions: Loading time of 3 min at 50 µL/min, linear gradient from 50 to 100% B in 15 min followed by an additional 5 min of 100% of B. 100 µL of the sample at a concentration of 10 µg/mL of each analyte were injected. The error bars represent the maximal and minimal values observed experimentally.”

3)     In Figure 1, each chemical structure of Figure 1A seems too small and Figure1B seems too wide. Therefore, Figure 1B should be adjusted to be a little narrow and Figure1A should be changed wider.

Response:

Thank you very much for this suggestion.  Figure 1 was modified as below.

4)     In Figure 2, as the context of the manuscript has said that the column can only be used for about 120 injections, but the legend of Figure 2 did not mention whether the authors used the same column or not. Therefore, the authors should add some word to explain it.

Response:

We appreciate your suggestion. If we understand correctly, there was a typing mistake in the question, been this related to figure 5. To address your suggestion, we modified the figure caption as presented below, been the information underlined added to it.

“Figure 5. Blanks were obtained after 50 and 100 analyses performed in the same column. Separation conditions: Loading time of 3 min at 50 µL/min, linear gradient from 50 to 100% B in 15 min followed by an additional 5 min of 100% B. No sample was injected.”

5)     Error bars appeared in both Figure 2 and Figure 3. However, the authors did not mention the replication number of relative experiments. The authors should explain it in the legend.

Response:

The error bars represent the maximum and minimal values obtained experimentally for each point; the figure caption was corrected as below.

“Figure 2. The area obtained with loading flow rates of 25, 50, and 100 µL/min (A) and % RSD for each loading flow rate (B) for pyriproxyfen and the area obtained with loading flow rates of 25, 50, and 100 µL/min (C) and % RSD for each loading flow rate (D) for abamectin B1a. Separation conditions: Loading time of 3 min at 50 µL/min, linear gradient from 50 to 100% B in 15 min followed by an additional 5 min of 100% of B. 100 µL of the sample at a concentration of 10 µg/mL of each analyte were injected. The error bars represent the maximal and minimal values observed experimentally.”

“Figure 3. The calibration curve (A) and % RSD for each point of the calibration curve (B) for pyriproxyfen and the calibration curve (C) and % RSD for each point of the calibration curve (D) for abamectin B1a.  Separation conditions: Loading time of 3 min at 50 µL/min, linear gradient from 50 to 100% B in 15 min followed by an additional 5 min of 100% of B. 100 µL of the sample were injected. The error bars represent the maximal and minimal values observed experimentally.”